# Proteomic Analysis of Brain Region and Sex-Specific Synaptic Protein Expression in the Adult Mouse Brain

**DOI:** 10.3390/cells9020313

**Published:** 2020-01-28

**Authors:** Ute Distler, Sven Schumann, Hans-Georg Kesseler, Rainer Pielot, Karl-Heinz Smalla, Malte Sielaff, Michael J Schmeisser, Stefan Tenzer

**Affiliations:** 1Institute for Immunology, University Medical Center of the Johannes Gutenberg-University Mainz, 55131 Mainz, Germany; malte.sielaff@uni-mainz.de; 2Focus Program Translational Neurosciences (FTN), University Medical Center of the Johannes Gutenberg-University Mainz, 55131 Mainz, Germany; 3Research Center for Immunotherapy (FZI), University Medical Center, Johannes Gutenberg-University Mainz, 55131 Mainz, Germany; 4Institute of Anatomy, Otto-von-Guericke University Magdeburg, 39120 Magdeburg, Germany; sven.schumann@med.ovgu.de (S.S.); hg.kesseler@gmx.de (H.-G.K.); 5Leibniz Institute for Neurobiology, 39118 Magdeburg, Germany; pielot@lin-magdeburg.de (R.P.); smalla@lin-magdeburg.de (K.-H.S.); 6Institute for Pharmacology and Toxicology, Otto-von-Guericke University Magdeburg, 39120 Magdeburg, Germany; 7Institute for Microscopic Anatomy and Neurobiology, University Medical Center of the Johannes Gutenberg-University Mainz, 55131 Mainz, Germany

**Keywords:** mass spectrometry-based proteomics, synapse, sex, hippocampus, striatum, prefrontal cortex, cerebellum, autism spectrum disorder (ASD), DDX3X, SET

## Abstract

Genetic disruption of synaptic proteins results in a whole variety of human neuropsychiatric disorders including intellectual disability, schizophrenia or autism spectrum disorder (ASD). In a wide range of these so-called synaptopathies a sex bias in prevalence and clinical course has been reported. Using an unbiased proteomic approach, we analyzed the proteome at the interaction site of the pre- and postsynaptic compartment, in the prefrontal cortex, hippocampus, striatum and cerebellum of male and female adult C57BL/6J mice. We were able to reveal a specific repertoire of synaptic proteins in different brain areas as it has been implied before. Additionally, we found a region-specific set of novel synaptic proteins differentially expressed between male and female individuals including the strong ASD candidates DDX3X, KMT2C, MYH10 and SET. Being the first comprehensive analysis of brain region-specific synaptic proteomes from male and female mice, our study provides crucial information on sex-specific differences in the molecular anatomy of the synapse. Our efforts should serve as a neurobiological framework to better understand the influence of sex on synapse biology in both health and disease.

## 1. Introduction

Synapses are the key structures for signal transduction and plasticity in the vertebrate central nervous system [1,2]. They form the core components of neural circuits and networks, collectively referred to as the brain connectome [3]. Although synapses were originally considered to be simple connection sites between neurons, the identification of synaptic proteins using mass spectrometry has transformed this view within the last decades [4,5,6,7,8,9,10,11]. Synapses are now recognized to be highly sophisticated computational units which are estimated to contain up to several thousands of different proteins [1,10,12]. Synapses are divided into two different subfamilies. While the family of electrical synapses is especially important during the development of the nervous system, chemical synapses are the dominant type of synapses in the adult nervous system [13,14]. Chemical synapses are composed of a transmitter-releasing presynaptic element and a signal-receiving and -processing postsynaptic compartment. The presynaptic compartment comprises hundreds of proteins centered around the vesicular neurotransmitter release machinery, which becomes active in response to an action potential into the presynaptic terminal [15]. The response to neurotransmitter release at the postsynaptic compartment is provided by a protein matrix of receptors and supporting proteins known as the postsynaptic density (PSD). The PSD has several functions including signal amplification, cytoskeletal anchorage, biochemical signaling modification and neurotransmitter receptor clustering [16,17]. Moreover, synapses of different regions of the brain are likely to have distinct protein compositions and thereby different functional properties [12,18,19,20,21].

Genetic disruption of synaptic proteins results in over 130 human neuropsychiatric disorders [22,23,24]. These so-called synaptopathies include complex disorders such as intellectual disability, schizophrenia and autism spectrum disorder (ASD) [25,26,27,28,29]. Animal mutant models of synaptopathies can mimic core aspects of the human diseases [30,31,32]. Despite all limitations, these models provide crucial information for the understanding of the biological pathways involved in the development of neuropsychiatric disorders. Additionally, pharmacological treatment of neuropsychiatric disorders often targets the synaptic interface [33,34,35]. Importantly, a wide range of synaptopathies display a sex bias in prevalence and clinical course [36,37,38]. Therefore, a detailed analysis of the synaptic proteome in different brain regions in males and females is required to understand the molecular basis of brain function and the etiology of its diseases. In this study we analyzed the synaptic proteome of the prefrontal cortex, hippocampus, striatum and cerebellum in male and female adult mice. Our efforts should serve as a neurobiological framework to better understand the molecular anatomy of synapses in a brain region and sex-specific manner both in health and disease.

## 2. Materials and Methods

### 2.1. Mice

Male and female 6-week-old C57BL/6J mice (P42) were used for this study. They were housed under defined conditions at a 12-h light/dark cycle and had free access to tap water and food. Mice were euthanized with carbon dioxide and the brain regions prefrontal cortex, hippocampus, striatum and cerebellum were dissected and stored at −80 °C after snap-freezing in liquid nitrogen. Animal experiments were performed in accordance with the regulations of the German Federal/Saxony-Anhalt State Law, the respective EU regulations, and the NIH guidelines. For each brain region, we generated four biological replicates of both, male and female mice. For each biological replicate material from three animals was pooled.

### 2.2. Subcellular Fractionation

For preparation of protein samples enriched for synaptic membrane structures, tissue was homogenized in 300 μL of 0.32 M sucrose with 5 mM HEPES and Complete™ protease inhibitor cocktail (Roche, Basel, Switzerland). Samples were centrifuged at 12,000 × *g* for 20 min. The resulting pellets were re-homogenized in 1 mL of 1 mM Tris/HCl, pH 8.1 containing protease inhibitors and incubated for 30 min at 4 °C. After incubation, samples were centrifuged at 100,000 × *g* for 1 h. The resulting pellets were re-homogenized in 400 μL of 0.32 M sucrose with 5 mM Tris/HCl, pH 8.1 and loaded on a 1.0 M/1.2 M sucrose step gradient. After centrifugation at 100,000 × *g* for 1.5 h synaptic membranes were collected at the 1.0 M/1.2 M sucrose interface. For proteome analysis, samples were resuspended in PBS and pelleted to reduce sucrose levels. A detailed description of the different enrichment steps is compiled in Appendix A. Moreover, by means of bioinformatic tools and immunoblot analysis we confirmed that our preparations are representative for synapse structures and synaptic substructures (Appendix A).

### 2.3. Proteolytic Digest of Enriched Synaptic Proteins

After enrichment, synaptic proteins were dissolved in a buffer containing 7 M urea, 2 M thiourea, 5 mM dithiothreitol (DTT), 2% (*w*/*v*) CHAPS and disrupted by sonication at 4 °C for 15 min using a Bioruptor (Diagenode, Liège, Belgium). The protein concentration was determined using the Pierce 660 nm protein assay (Thermo Fisher Scientific, Waltham, MA, USA) according to the manufacturer’s protocol. 20 µg of total protein were subjected to tryptic digestion using a modified Filter Aided Sample Preparation (FASP) protocol as described in detail before [39,40]. In brief, proteins were transferred onto spin filter columns (Nanosep centrifugal devices with Omega membrane, 30 kDa MWCO; Pall, Port Washington, NY, USA) and detergents were removed washing the samples three times with a buffer containing 8 M urea. Proteins were reduced using DTT and alkylated with iodoacetamide (IAA). Afterwards, excess IAA was quenched with DTT and the membrane was washed three times with 50 mM NH_4_HCO_3_ followed by overnight digestion at 37 °C with trypsin (Trypsin Gold, Promega, Madison, WI, USA). An enzyme-to-protein ratio of 1:50 (*w*/*w*) was used to digest the proteins. After digestion, peptides were recovered by centrifugation and two additional washes with 50 mM NH_4_HCO_3_. After combining the flow-throughs, samples were acidified with trifluoroacetic acid (TFA) to a final concentration of 1% (*v*/*v*) TFA and lyophilized. Purified peptides were reconstituted in 0.1% (*v*/*v*) formic acid (FA) for LC-MS analysis.

### 2.4. Nanoscale Liquid Chromatography Mass Spectrometry (nanoLC-MS) Analysis

Samples were analyzed by LC-MS using a Synapt G2-S HDMS mass spectrometer (Waters Corporation, Milford, MA, USA) coupled to a nanoAcquity UPLC system (Waters Corporation, Milford, MA, USA). Water containing 0.1% (*v*/*v*) FA, 3% (*v*/*v*) dimethyl sulfoxide (DMSO) was used as mobile phase A and acetonitrile (ACN) containing 0.1% FA (*v*/*v*), 3% (*v*/*v*) DMSO as mobile phase B [41]. Tryptic peptides (corresponding to 200 ng) were loaded onto an HSS-T3 C18 1.8 μm, 75 μm × 250 mm reverse-phase column from Waters Corporation in direct injection mode and were separated running a gradient from 5–40% (*v*/*v*) mobile phase B over 90 min at a flow rate of 300 nL/min. After separation of peptides, the column was rinsed with 90% mobile phase B and re-equilibrated to initial conditions resulting in a total analysis time of 120 min. The column was heated to 55 °C. Eluting peptides were analyzed in positive mode ESI-MS by ion-mobility separation (IMS) enhanced data-independent acquisition (DIA) UDMS^E^ mode as described in detail before [40,42]. Acquired MS data were post-acquisition lock mass corrected using [Glu1]-Fibrinopeptide B, which was sampled every 30 s into the mass spectrometer via the reference sprayer of the NanoLockSpray source at a concentration of 250 fmol/µL. All samples (i.e., biological replicates) were analyzed by LC-MS in duplicates. Moreover, to monitor reproducibility and long-term stability of the LC-MS platform, we generated four sample pools, one for each brain region. Toward this end, equal amounts of the four female and four male biological replicates were mixed for each brain region. LC-MS analyses of the sample pools were scheduled between the actual sample runs resulting in up to six replicate measurements for the sample pools.

### 2.5. Data Processing and Label-Free Quantification Analysis

Raw data processing and database search of LC-MS data were performed using ProteinLynx Global Server (PLGS, ver.3.0.2, Waters Corporation, Milford, MA, USA). Data were searched against a custom compiled UniProt mouse database (UniProtKB release 2018_09, 16,991 entries) that contained a list of common contaminants. The following parameters were applied for database search: (i) trypsin as enzyme for digestion, (ii) up to two missed cleavages per peptide, (iii) carbamidomethyl cysteine as fixed, (iv) and methionine oxidation as variable modification. The false discovery rate (FDR) for peptide and protein identification was assessed using the target-decoy strategy by searching a reverse database. FDR was set to 0.01 for database search in PLGS.

Post-processing of data including retention time alignment, exact mass retention time as well as IMS clustering, normalization and protein homology filtering was performed using the software tool ISOQuant ver.1.8 [40,42]. Algorithms and ISOQuant settings have been described in detail before [40,42]. For cluster annotation in ISOQuant, an experiment-wide FDR of 0.01 was applied at the peptide-level. To be included in the final list a peptide had to be identified at least four times across all runs. Only proteins that had been identified by at least two peptides with a minimum length of seven amino acids, a minimum PLGS score of 6.0 and no missed cleavages were used for quantification and included in the final dataset. For each protein absolute in-sample amounts were calculated using TOP3 quantification as described before [43]. The mass spectrometry proteomics data have been deposited to the ProteomeXchange Consortium (http://proteomecentral.proteomexchange.org) via the PRIDE partner repository [44] with the dataset identifier PXD015610.

Statistical analysis of the data was conducted using Student’s *t*-test, which was corrected by the Benjamini–Hochberg (BH) method for multiple hypothesis testing (FDR of 0.05). T-tests were only calculated if a protein was identified at least in three biological replicates. R (version 3.6.1) was used for further analyses and to plot the data [45,46,47,48]. Functional annotation analysis of synaptic proteins that displayed significant changes between brain regions (after BH correction, log_2_ fold change >1) was performed using the Gene Ontology (GO) knowledgebase (http://geneontology.org/) [49,50].

## 3. Results

### 3.1. Differential Expression of Synaptic Proteins across Different Brain Regions

To resolve the brain region and sex-specific mouse synaptic proteome, we enriched pre- and post-synaptic proteins from the (i) prefrontal cortex, (ii) hippocampus, (iii) striatum, and (iv) cerebellum of adult mice (P42). In total, four biological replicates (each pooled from three mice) of both, male and female animals, were collected for each brain region. Synaptic proteome samples were analyzed after tryptic digestion by DIA LC-MS (Figure 1a). Combined label-free quantification analysis of all replicates revealed distinct, brain region-specific synaptic protein expression patterns (Figure 1b). Around 3000 proteins could be quantified in each brain region, with a total of 3173 proteins (corresponding to over 40,000 peptides) in the complete dataset (Appendix A). Out of the 3173 proteins, 2896 proteins were identified in all four brain regions (Figure 2a). To assess the quality of our proteome analysis, we quantitatively compared protein abundances across the whole dataset. Between biological replicates, Pearson’s correlation coefficients for protein abundances were between 0.86 and 0.99, respectively, demonstrating high technical and biological reproducibility (Appendix A).

Almost all proteins in the present dataset showed brain region-dependent expression profiles and were significantly enriched in either a single or two brain regions (Figure 1b and Figure 2b and Appendix A). Only a small subset of proteins (a total of 142) could not be assigned to distinct brain regions. Some of these proteins did not pass our filter criteria (i.e., were present in less than three biological replicates in the respective brain region(s), variation between biological replicates was too high) or were enriched in three brain regions, i.e., displayed lower expression in a single region. In total, 24 proteins showed stable and similar protein levels across all replicates (i.e., were identified in all runs with a coefficient of variation (CV) for the protein abundance < 25%).

Hierarchical clustering indicates that the cerebellum is the most diverging region, whereas the synaptic proteomes of the prefrontal cortex and hippocampus show the highest similarities (Figure 1b). We detected 650 proteins that were significantly enriched in the cerebellum as compared to the other brain regions, followed by the striatum with 490 region-specific proteins (Figure 2b). In case of the prefrontal cortex and the hippocampus, the number of enriched proteins was markedly lower (143 and 182, respectively). Moreover, in line with previous findings [51], cortical and hippocampal synapses share the most proteins with similar expression patterns (Figure 2b, Appendix A).

Proteins significantly enriched in striatal and cortical synapses with more than a twofold expression difference as compared to other regions were mainly associated with mitochondria and the cytoplasm (Figure 2c). In case of striatum, our analyses additionally revealed a high enrichment of neuronal and synaptic proteins, including voltage-gated potassium channels, Ras family members as well as receptor tyrosine and MAP kinases (Figure 2c). The top 15 biological processes associated with striatal-specific synaptic proteins mainly relate to mitochondrial processes and functions (Figure 2d). However, we also found a high enrichment for proteins involved in dopamine signaling and exocytosis such as the D(1A) dopamine receptor (DRD1), the sodium-dependent dopamine transporter (SC6A3) or Vacuolar protein sorting-associated protein 11 homolog (VPS11), which is required for the fusion of endosomes and autophagosomes with lysosomes (Appendix A). In case of the prefrontal cortex, no biological process was significantly enriched. However, among the cortex-enriched synaptic proteins, we detected, for example, the neuronal migration protein doublecortin (DCX), which is involved in the initial steps of neuronal dispersion and cortex lamination during cerebral cortex development [52]. Other proteins are associated with mitochondrial functions or display protein serine/threonine kinase activity such as the cation channel TRPM6 or the death-associated protein kinase 1 (DAPK1), which is involved in multiple cellular signaling pathways triggering cell survival, apoptosis, and autophagy.

GO enrichment analysis for hippocampal and cerebellar synaptic proteins highlighted the (glutamatergic) synapse, the cell junction and the (intracellular) organelle part as the most significant components, respectively (Figure 2c). Proteins exclusively identified in the cerebellum (Figure 2a) include, for example, the GABA(A) receptor subunit alpha-6 (GBRA6) involved in GABAergic synaptic transmission, the Purkinje cell protein 2 (PCP2) and Cerebellin-1 (CBLN1). The cerebellum-specific protein CBLN1 is involved in cerebellar granule cell differentiation and essential for cerebellar synaptic integrity and plasticity [53]. Downregulation or loss of CBLN1, a key node in the protein interaction network of ASD genes, impairs sociability and weakens glutamatergic transmission in ventral tegmental area (VTA) neurons [54]. Moreover, GO analysis of biological processes revealed an enrichment of proteins at the cerebellar synapse that are associated with mRNA processing/splicing (Figure 2c and Appendix A). Alternative splicing is a crucial mechanism for neuronal development, maturation, as well as synaptic properties [55] and local protein synthesis is a ubiquitous feature of neuronal pre- and post-synaptic compartments [56]. Regarding the hippocampus, GO analysis of biological processes revealed that proteins involved in neurogenesis and cell differentiation are enriched at its synapse (Figure 2c and Appendix A). Moreover, proteins involved with typical hippocampal functions include, for example, the sodium/calcium exchanger 2 (NAC2), which is essential for the control of synaptic plasticity and cognition [57], or the protein-tyrosine kinase 2-beta (FAK2), which is associated with long-term synaptic potentiation and depression.

### 3.2. Sex-Specific Differences in the Synaptic Proteome

One major focus of the present study was to resolve sex-specific differences in the synaptic proteome across different brain regions of adult mice (Figure 3).

We observed the highest divergency between male and female mice in the hippocampus (Figure 3a,b). In total, 71 proteins showed differences in their expression levels between the two sexes including multiple proteins known to be involved in neurological disorders (such as Parkinson’s and Alzheimer’s disease) (Appendix A). Only little differences between male and female mice were observed in the striatal and the cortical synaptic proteome. Here, only seven and eight proteins differed significantly in their abundance, respectively. In the cerebellum, we detected 28 differentially expressed proteins comparing male and female animals, mainly involved in neuron projection and synaptic transmission, as well as in RNA binding and processing (Appendix A). In a recent study, Block et al. [58] investigated sex differences in protein expression for a selected panel of about 100 proteins associated with learning/memory and synaptic plasticity in the hippocampus, cerebellum, and cortex of female and male controls and their trisomic littermates (Dp(10)1Yey mouse model of down syndrome). In line with our findings, the authors observed by far the most differences in the hippocampus between the two sexes in their control group, followed by the cerebellum.

Interestingly, we observed no overlap of sex-associated synaptic proteins between the different brain regions. Only one protein displayed differential expression across all regions between male and female mice, the ATP-dependent RNA helicase DDX3Y (Figure 3b). As the Ddx3y gene is located on the chromosome Y, it is expected that the respective gene product will be only found in male individuals. Interestingly, its paralog DDX3X, is listed as strong ASD candidate (category 2) in the SFARI autism gene database and has been associated with cases of intellectual disability, hyperactivity, and aggression in females [59]. Hence, we compared the quantitative datasets of altered proteins between male and female wildtype animals with selected autism-associated target genes. In total, we selected 257 ASD risk genes (196 after filtering for duplicates and excluding those without homologues in mouse) for the comparison. Selected ASD risk candidates were compiled from three sources: i) the SFARI autism gene database, the studies from ii) Rubeis et al. [60] and iii) Doan et al. [61] (Appendix A). Regarding the SFARI gene set, we included high confidence (category 1) and strong ASD candidates (category 2) comprising 25 and 66 candidates, respectively. From the study of Rubeis et al. we incorporated the set of 107 autosomal ASD risk genes (FDR < 0.3) [60] and from Doan et al. 41 recessive genes specifically knocked out (i.e., carrying biallelic loss-of-function (LOF) mutations) in individuals diagnosed with ASD as well as 18 genes detected in their ASD cohort either with LOF or biallelic, damaging missense mutations that have been already described as pathogenic or likely pathogenic [61]. Up to 70 gene products of the described ASD risk genes were detected in our dataset (including 21 that have been described by multiple studies; Appendix A). Out of these, four proteins were found to be differentially expressed at the synapses of male and female mice including DDX3X (Figure 3d, Appendix A) as well as KMT2C, MYH10 and SET (Appendix A).

## 4. Discussion

The major scope of the present study was to resolve sex-specific differences in the mouse synaptic proteome across different brain regions in adult mice at a postnatal age of P42. It has been nicely shown by Gonzalez-Lozano et al. [6] for cortical mouse synapses that levels of synaptic proteins generally increase throughout brain development and converge at an adult age, whereas other proteins, e.g., involved in protein synthesis, are more likely to decrease in abundance during maturation. At P42 typical pre- and post-synaptic proteins already display stable expression levels as compared to later timepoints thus adequately representing the adult mouse synaptic protein repertoire. In general, proteomic studies on brain samples show a great variability in sample preparation [1,6,9,10,11,21] leading to difficulties in direct comparability. In this study, we enriched proteins from both, the pre- and postsynaptic compartment of the synapse, thereby giving an upmost comprehensive view on the proteinaceous inventory of the synaptic interface. Our unbiased proteomic approach is therefore capable to identify novel sex-specific molecular targets in male and female synapses, respectively. Despite the aforementioned difficulties in comparability, our data are in line with findings of other recent proteome studies on the nervous system. Our data, for example, strongly support the findings by Mann et al. [51] and Alvares-Castelao et al. [21] that the proteome of cerebellar neurons, is highly diverging from cortical, hippocampal and striatal neuron proteomes. In 2015, Sharma et al. [51] resolved a brain region and cell-type specific mouse brain proteome. Analyzing complete brain regions without subcellular enrichment, they report highest divergence for the cerebellum (along with the optic nerve and the brain stem). Among the 10 analyzed brain regions in their study, hippocampus, striatum, prefrontal and motor cortex showed highest similarities. This is highly comprehensible due to the different ontogenetic and phylogenetic development of the rhombencephalic cerebellum and the prosencephalic cortex, hippocampus and striatum. Proteins that showed brain region-dependent expression differences in the dataset of Sharma et al. were associated with the (post)synaptic membrane and involved in processes like transmembrane transporter activity and synaptic transmission underlining the importance to further resolve the synaptic protein composition to better understand underlying neurological and synaptic processes across different regions of the brain.

On the synaptic level, only a limited amount of morphological differences between sexes have been reported, yet. In human, Alsonso-Nanclares et al. [62] found that men have a significantly higher density of synaptic contacts than women in all cortical layers of the temporal neocortex. In rodents, it was shown that the density of dendritic spines in the hippocampal CA1 region and the nucleus accumbens is higher in females [63,64]. Importantly, dendritic spine density is influenced by the estrus cycle in rodents [65]. It is well known that sex steroid hormones have an impact on synaptic function and synaptogenesis/synaptic plasticity in a sex-specific way [66,67,68,69,70]. Expression and subcellular localization of nuclear and membrane-associated steroid hormone receptors is different in male and female neurons thereby leading to different responses on hormone action. Impressively, there is no evidence for a lack of steroid hormone receptors in any brain region [71]. In the present study, we actually found no significant differences in the expression of the classical and putative membrane-associated steroid hormone receptors in the synapses of either brain region.

It has further been shown that calcium/calmodulin kinase kinase (CaMKK) signaling differs in male and female mice [72]. This is in accordance with our finding of a sex-specific difference of synaptic calcium/calmodulin dependent protein kinase II delta (KCC2D) levels in the hippocampus. Zettergren et al. found that myristoylated alanine rich C kinase substrate (MARCKS) protein, a cellular substrate for protein kinase C is more highly expressed in neurons of the limbic system (hypothalamus/amygdala) of neonatal female mice compared to male littermates [73]. In contrast to our study, no isolation of subcellular fractions and comparison of different brain regions was performed. Moreover, we could not find a significant difference in the synaptic amount of MARCKS between male and female. This difference could be explained by our focus on the synaptic compartment or the adult age of the animals analyzed.

In our synaptic proteome dataset, we could identify sex-specific molecular changes in all brain regions analyzed. Curiously, only the Y chromosome-encoded DDX3Y protein was differentially expressed in all four regions. In mice, DDX3Y is expressed in several tissues including the brain [74]. In contrast, in humans DDX3Y is an important regulator of spermatogenesis exclusively expressed in human testis [75]. Because of the y-chromosomal heritage, the absence of DDX3Y in female brain was reasonable. Interestingly, the DDX3Y paralog DDX3X shows a significant higher expression only in the female striatum compared to male. DDX3X is a multifunctional ATP-dependent RNA helicase. Although its exact physiological function in the organism is still not fully understood, it seems to be involved in multiple steps of gene expression, such as transcription, mRNA maturation and translation. *DDX3X* is listed as strong ASD candidate (category 2) in the SFARI autism gene database. ASD is a heterogeneous group of neurodevelopmental disorders, characterized by early-onset deficits in social interaction and communication skills, together with restricted, repetitive behavior. Defects in DDX3X function in humans is associated with brain and behavioral abnormalities, microcephaly, facial dysmorphism, hypotonia, aggression and movement disorders and/or spasticity in female and probably in male [59,76,77,78,79,80,81,82,83,84,85]. The finding of a sexual dimorphic autism related protein specifically in the striatum is of particular interest because defects in striatal circuitry are known to cause autism-like phenotypes [86]. Interestingly, a sexually dimorphic phenotype has further been observed in a mouse model of striatal interneuron depletion [87]. Another autism related protein, the histone methyltransferase KMT2C was found to be reduced in the hippocampal synapse of male mice. In humans, a mutation of *KMT2C* is associated with a clinical phenotype overlapping Kleefstra syndrome [88]. Also, the murine variant of the non-muscle heavy chain II B, encoded by the Myosin Heavy Chain 10 gene (*MYH10*) was found to be less expressed in the synapse from the male hippocampus. In humans, mutation of *MYH10* leads to a severe CNS phenotype characterized by microcephaly, cerebral and cerebellar atrophy and severe intellectual disability [89]. The gene encoding the protein SET, which showed increased expression level in the cortical synapse in female mice, is listed as strong ASD candidate (category 2) in the SFARI autism gene database. The multitasking protein SET is a nuclear proto-oncogene [90] and involved in apoptosis [91], transcription, nucleosome assembly and histone chaperoning [92]. SET inhibits acetylation of nucleosomes, especially histone H4, by histone acetylases (HAT) [93]. This inhibition is most likely accomplished by masking histone lysines from being acetylated, and the consequence is to silence HAT-dependent transcription. Mutations in the gene encoding SET are linked to developmental delay and intellectual disabilities as well as to autosomal dominant 58 (MRD58), a form of mental retardation, characterized by significantly below average general intellectual functioning associated with delayed development, impairments in adaptive behavior, language delay and speech impairment [94,95,96]. Interestingly, SET interacts with intracellular domains of the gonadotropin-releasing hormone (GnRH) receptor and differentially regulates receptor signaling to cAMP and calcium in gonadotrope cells [97]. Notably, a recent study showed that SET expression is regulated by the neurohormone GnRH [98], providing a potential molecular basis for sex-specific differences in expression levels.

Despite our findings, several questions remain to be addressed in future studies. Efforts in recent years have been made to resolve the spatial distribution of synapse types and subtypes [20,21] as well as to decipher the protein repertoire of excitatory and inhibitory synapses [9]. Although we could identify brain region-specific synaptic proteins that are differentially expressed in the synapses of male and female mice, it remains subject to future analyses to link sex-specific expression patterns to specific synapse subtypes or sublocations to gather further insights into the sex-related physiology of neuronal function. Moreover, an age-dependent analysis would further improve our understanding of sex-specific differences during neuronal development.

## 5. Conclusions

Taken together, our work reveals the first sex-specific synaptic proteome in mice. First, we were able to confirm former findings of a specific repertoire of synaptic proteins in different brain areas. Second, we found a set of novel proteins differentially expressed in the synapses of males and females, respectively. Importantly, the repertoire of sex-specific expressed proteins is also brain region-specific. Our findings reveal novel insights into the sex-specific differentiation of synapses thereby leading to a better understanding of the sex-specific physiology of neuronal function and behavior and the pathophysiology of neurodevelopmental and neuropsychiatric diseases in general that often carry a so-called sex bias.

## Figures and Tables

**Figure 1 cells-09-00313-f001:**
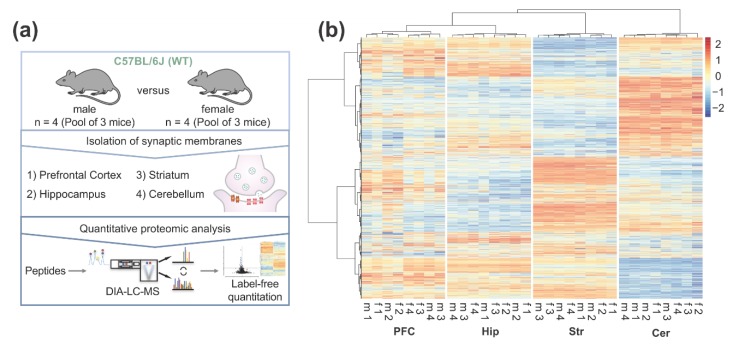
Quantitative LC-MS analysis of the brain region-specific synaptic proteome in male (m) and female (f) adult mice: (**a**) Graphical illustration of the workflow for the characterization of the brain region- and sex-resolved synaptic proteome; (**b**) Heatmap of all quantified proteins in the dataset. For cluster analysis and heatmap visualization, label-free quantification values were log_2_-transformed and scaled subtracting the mean of the row followed by the division of resulting values by the standard deviation of the row.

**Figure 2 cells-09-00313-f002:**
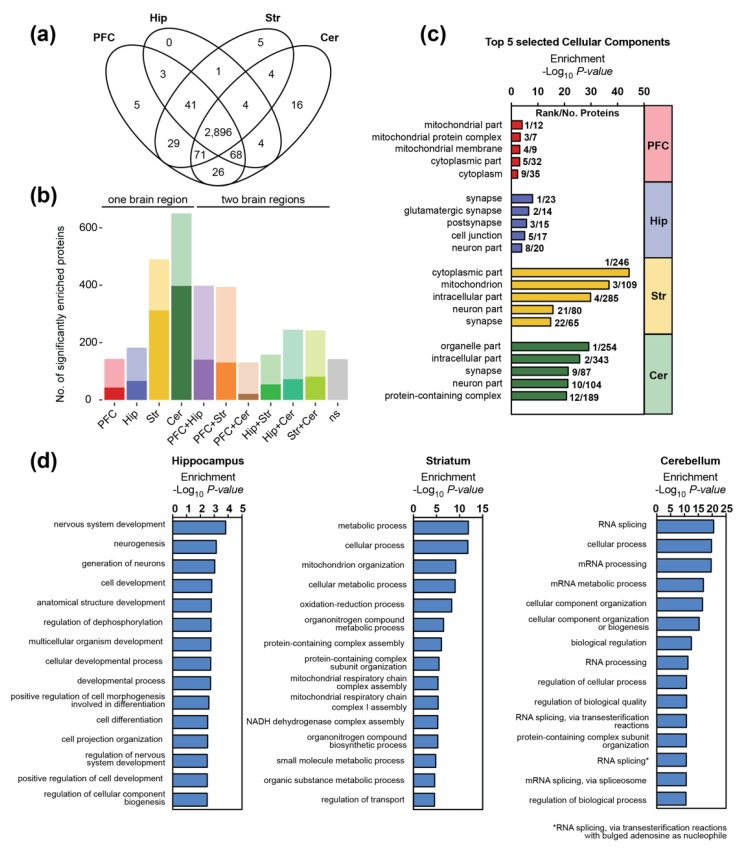
Composition of the synaptic proteome differs between brain regions: (**a**) Overlap of proteins identified at the synapses of the prefrontal cortex (PFC), hippocampus (Hip), striatum (Str) and cerebellum (Cer). Presence of the proteins was inferred following alignment between runs. (**b**) Number of significantly enriched proteins in one or two brain regions (BH corrected Student’s *t*-test, *p* < 0.05). Proteins were always assigned to the group displaying the highest significance (i.e., lowest p-value). Transparent bars display numbers of all significant proteins and non-transparent bars proteins that are at least 2-fold enriched; (**c**,**d**) Gene Ontology (GO) enrichment analysis of synaptic proteins that are significantly associated with a certain brain region (Benjamini–Hochberg correction, *p* < 0.05, log_2_ fold change compared to other regions >1). (**c**) Selected GO terms for components as well as the (**d**) top 15 biological processes are displayed. In case of PFC-specific synaptic proteins, no biological process was significantly enriched.

**Figure 3 cells-09-00313-f003:**
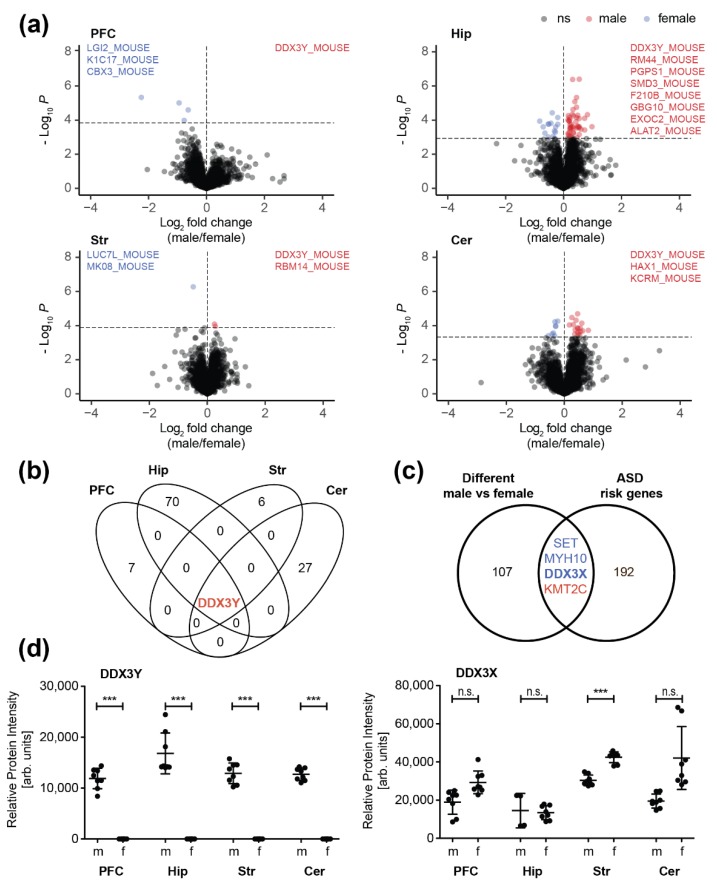
Sex-related differences in synaptic proteome composition: (**a**) Volcano plots displaying differentially regulated synaptic proteins of the prefrontal cortex (PFC), hippocampus (Hip), striatum (Str), and cerebellum (Cer) between male (m) and female (f) mice. Significantly regulated proteins (BH corrected Student’s *t*-test, *p* < 0.05) are highlighted blue (female) and red (male). Uniprot entries listed in the plots mark proteins, exclusively detected either in male (blue) or female mice (red). (**b**) Overlap of significantly regulated proteins (male versus female) in different brain regions. (**c**) Overlap of all differently expressed proteins between male and female mice with autism spectrum disorder (ASD) risk genes (see Appendix A); (**d**) Relative protein levels of DDX3Y and DDX3X. Asterisks (***) indicate highly significant differences in protein abundances between the two sexes (*p* < 0.001, Student’s *t*-test). n.s., not significant.

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
