# Peer review of "Proteomic Analysis of Brain Region and Sex-Specific Synaptic Protein Expression in the Adult Mouse Brain"

_cells, 2020, doi:10.3390/cells9020313_

Round 1

Reviewer 1 Report

this is a very well written paper looking at different brain regions and sex differences in the mouse brain.

i have a few comments before i think the paper is ready for approval.

page 2, mice: the research ethics committee number under which approval was granted needs to be added. how many animals per group were used for this study (m/f)? this would be important to state here already.

this is a fractionation method of pre-and postsynaptic densities, i would prefer to harmonise thenaming as there seems to be a lot of differetn names out there in the literature, this should be called a synaptosomal fraction (e.g. see work by Bayes et al, or Hahne C.G. et al)

page 3, line121/122: biological replicates from each brain region were pooled. could the authors please elaborate on this? how many brains were pooled, how many pools were used, does the 6 replicate measurements refer to biological or technical replicates?

page 5, line 193: a coefficient of variation of 25% seems extremely high, especially if pools of mice were used, could the authors add more details to this or should they change their calculations to allow a CV that is much lower?

the results are very detailed and confirm and confirm previous findings and add new ones, nevertheless, the citing of other studies in the results is a mix of results and discussion and i therefore suggest, that the authors highlight this by combining results and discussion as one part of the manuscript.

the authors do not state any limitations to the study, are they really none?

the biggest surprise to me was that the authors seem to not compare their work to work of Bayes and Grant, e.g. Nature Neurosci. 2012).This group has a very detailed list of mouse synaptic proteins on their site(http://synaptome.genes2cognition.org/) that would strengthen the current findings quite a bit.

Reviewer 2 Report

A brief summary (one short paragraph) outlining the aim of the paper and its main contributions.

In manuscript cells-618595, Distler et al. perform a label-free differential protein expression analysis on the synaptic proteomes of different brain regions from male and female mice. They interpret the resulting protein lists in light of human neuropsychiatric disorders with sex differences. The paper is well written and clear and is a relevant contribution to the field. However, the issues below need to be addressed before it is suited for publication.

Broad comments highlighting areas of strength and weakness. These comments should be specific enough for authors to be able to respond.

The authors conducted the LC-MS analysis according to the prevailing standards and the data can be expected to be qualitative. They assessed the reproducibility of their analysis as well, with the inclusion of quality control samples to avoid batch effects from instrumental decline.

The main concern however, is that the authors are too careless on their claim about the synaptic origin of the proteins that are measured. More specifically, the purity of the synaptic fraction is not validated, while the authors state “In the synaptic junctions of each brain region at least 3,000 proteins could be quantified” and they do use “synaptic proteomes (P5L239)” as a term to describe their samples. Instead, they asses enrichment by sucrose gradient ultracentrifugation (which is not a purification) using only one marker (PSD-95) on western blot. Additionally, the authors do not refer to any other study that has used a similar enrichment protocol and wherein the approach is more thoroughly validated (see minor remarks). Still, they recognize in the discussion that “In general, proteomic studies on brain samples show a great variability in sample preparation [1,6,9–11,21] leading to difficulties in direct comparability.”

While proteomics experts usually restrain from bold conclusions when they look at such protein lists, other molecular biologists are less aware of the dangers and they are given the impression that indeed these 3000 proteins all are synaptic proteins. Based only on sucrose gradient ultracentrifugation, this is practically unimaginable. These protein lists contain so many proteins that they become tools for confirmation bias of future experiments. The authors should make the distinction between enrichment and purification more clearly in the text and should more critically assess the GO terms that are given. More specifically, it would have been more useful to do a GO (enrichment) analysis on the complete homogenate and the S2 fraction as well, because most likely “synapse” will be a top hit in all these brain fractions. Even more useful would be to have a list of proteins known to be absent in synapses (e.g. nuclear proteins) and verify their abundance in the current dataset (as is done by Bayes et al., ref 22).

Overall, I would argue that the focus should be on the differential analysis (figure 3) rather than on the functional analysis of the protein list (Figure 2), to avoid future misuse of the protein list. This could be done by e.g. moving figure 2 to supplementary results, after the important nuances are added to the text. Any experiment/result that more clearly describes the purity of the synaptic proteome would greatly improve the quality of the manuscript and would justify the main figure 2.

Specific comments referring to line numbers, tables or figures. Reviewers need not comment on formatting issues that do not obscure the meaning of the paper, as these will be addressed by editors.

P2L80: Was the isolation protocol based on a previously reported and more thoroughly validated protocol? If so, please refer to it.

P3L111: The sample loading is not specified. Please mention how much of the tryptic peptides were loaded on-column.

P3L123: Please add whether the samples were randomized.

P3L136: please clarify what is meant by “cluster annotation in ISOQuant”. Also, while requiring the recurrence of the same identification does control the experiment-wide FDR to some extent, this is not quantifiable. Indeed, an experiment-wide FDR control would be a very interesting validation of this strategy. By exporting all identifications of the entire experiment (both targets and decoys) and sorting these by score, an experiment-wide 1% FDR can easily be set in the final list. There is no need to redesign all figures, but maybe to add a supplementary figure/table that makes the comparison between these two approaches. This is a measure to further reduce over-reporting of proteins as part of the synaptic proteome.

P4L184: please clarify what is meant by “standardized”.

Figure 2a: P3L133 mentions that an alignment between runs was performed. If that is true, a Venn Diagram is a misleading representation for protein presence. More specifically, the legend states “proteins identified in…” and this should be nuanced to “presence of the proteins was inferred following alignment between runs”, or alike.

Figure2c-d: The GO analyses are not very informative in their current form. Are mitochondria known to be more enriched in striatal and cortical synaptic junctions? Or are these less pure fractions? Especially without a broader reference of the complete brain homogenate, these terms are impossible to interpret. Either way, as mentioned under major remarks, I believe that the emphasis should be on the results from figure 3 and on specific interesting proteins that are differential, because these are unrelated to the purity of the “synaptic proteome”.    

P6L245: Ref 51 should be Sharma et al.

P6L252: the authors probably intended to say “underlining”.

Reviewer 3 Report

The manuscript by Ute Distler et al. represents a good example of a focused study on synaptic protein expression, the cellular compartment which is often targeted in many neurodegenerative and mood disorders, including schizophrenia and autism spectrum disorder (ASD).

The authors have analyzed the proteome of synaptic junctions in the prefrontal cortex, hippocampus, striatum and cerebellum of male and female adult C57BL/6J mice, revealing region-specific differentially expressed proteins between male and female individuals. Interestingly, the putative markers DDX3X, KMT2C, MYH10 and SET of ASD were pinpointed.

While, overall, the study is quite interesting and well presented it lacks solid validation by independent set of techniques. It would be of high interest to the readers to demonstrate whether the detected differences can indeed by shown and at best what is the functional basis for such changes. Therefore, some examples of independent validation are required to consider the study of sufficient quality for publication in Cells.

Round 2

Reviewer 3 Report

The authors have adequately resolved majority of issues and introduced changes required by the reviewer. Their effort to resolve the validation of the differential protein expression data by quantitative Western blotting with well-characterized antibodies is adequately documented. One issue still remains, however, concerning the quality of presentation of one example of the Western blot provided by the authors. It would be of high interest to provide the appropriate loading control as well as statistics to truly appreciate the direction of protein expression changes. Such figure shall be shown as a supplementary data and the results mentioned in the text.
